# Two-Dimensional Conjugated Mass Transfer of Carbon Dioxide Absorption in a Hollow-Fiber Gas-Liquid Membrane Contactor

**DOI:** 10.3390/membranes12101021

**Published:** 2022-10-20

**Authors:** Chii-Dong Ho, Luke Chen, Chien-Chang Huang, Chien-Hua Chen, Thiam Leng Chew, Yu-Han Chen

**Affiliations:** 1Department of Chemical and Materials Engineering, Tamkang University, New Taipei City 251301, Taiwan; 2Department of Water Resources and Environmental Engineering, Tamkang University, New Taipei City 251301, Taiwan; 3Department of Chemical and Materials Engineering, National Ilan University, Yilan County 260007, Taiwan; 4Department of Chemical Engineering, Faculty of Engineering, Universiti Teknologi PETRONAS, Seri Iskandar 32610, Malaysia; 5CO2 Research Center (CO2RES), Institute of Contaminant Management, Universiti Teknologi PETRONAS, Seri Iskandar 32610, Malaysia

**Keywords:** carbon dioxide absorption, MEA absorbent, hollow-fiber membrane contactor, conjugated Graetz problem, Sherwood number

## Abstract

The absorption efficiencies of CO_2_ in hollow-fiber membrane contactors using an ethanolamine (MEA) solvent under both concurrent- and countercurrent-flow operations were investigated theoretically and experimentally. Two-dimensional mathematical modeling was developed by Happel’s free surface model, and the resultant partial differential equations were solved analytically using the separated variables method with the use of an orthogonal expansion technique. A simplified expression of Sherwood number variations was reported by employing the relevant operations conditions and expressed in terms of the computed eigenvalues for predicting concentration distribution and absorption efficiency. It is emphasized that, in comparing various fiber packing configurations, both theoretical predictions and experimental results should be compared to find the absorption flux increment accomplished by the CO_2_/N_2_ stream passing through the fiber cells under the same mass flow rate. The value of the present mathematical treatment is evident to propose a simplified expression of the averaged Sherwood number variations, and provides the predictions of the absorption flux, absorption efficiency, average Sherwood number with the absorbent Graetz number, inlet CO_2_ concentration, and absorbent flow rates as parameters. The availability of such concise expressions, as developed directly from the analytical formulations, is the value of the present study. The experiments of the CO_2_ absorption using MEA with alumina (Al_2_O_3_) hollow fiber membranes are also set up to confirm the accuracy of the theoretical predictions. The accuracy derivations between the experimental results and theoretical predictions for concurrent- and countercurrent-flow operations are $4.10\times 10^{-2}\leq E\leq 1.50\times 10^{-2}$ and $1.40\times 10^{-2}\leq E\leq 9.0\times 10^{-1}$, respectively. The operations of the hollow-fiber membrane contactor implementing *N* = 7 fiber cells and *N* = 19 fiber cells offer an inexpensive method of improving absorption efficiency by increasing fiber numbers with consideration of device performance.

## 1. Introduction

The application of membrane technology to physical/chemical gas absorption is the most common purification technology for gas separation in removing CO_2_ from fossil fuel combustion to reduce greenhouse gas emissions [1]. Membrane technology has been extensively applied to liquid/liquid and gas/liquid systems and is widely used in many separation processes, such as gas absorption and metal ion removal [2], due to the advantages of lower energy consumption [3], a larger mass transfer area, continuous operations, and the flexibility to scale up [4]. Some membrane materials, such as PMSQ (polymethylsilsesquioxane) and hybrid silica aerogel, were used to be durable and reusable to enhance the CO_2_ absorption flux considerably [5,6]. The membrane is in contact with gas/liquid on each side, as solvent absorption is operated with a microporous membrane module. The distribution coefficient of gas solute in the two-phase system existing a gradient composition in between plays an important role in the separation efficiency of the membrane of physical absorption. Chemical absorbents of monoethanolamine (MEA) solutions on the membrane surface of the liquid phase in the hydrophobic microporous membrane module as a more efficient absorption process could enhance CO_2_ being selectively absorbed. Currently, chemical absorption by amine solutions is the most advanced technology for the mechanisms of CO_2_ absorption from gas mixtures, as confirmed by a previous study [7,8]. The advantage of chemical absorption technology is that it has been commercialized for many decades with various amines and mixed amines [9] used widely to enhance CO_2_ capture efficiency and reduce regeneration costs [10]. The most commonly used hollow fiber membrane contactors were designed in a shell/tube configuration with the shell side (absorbent) parallel to the fiber cells (CO_2_), which were either in concurrent- or countercurrent-flow operations. The microporous hydrophobic membrane device acts as a gas absorber, while the amine solution flowing on the other side directly contacts the membrane surface. Rongwong et al. [11] provided a better understanding of membrane gas-absorption operations than conventional gas-absorption processes, and the separation efficiency was determined on the distribution coefficient of gas solutes in both gas and liquid phases [12]. The advantages of combining chemical absorption and membrane separation techniques in membrane absorption modules with a higher specific area offer selective absorption of the desired gas mixture component [13]. The alkanolamine-based CO_2_ absorption processes have been used widely and commercially. Successful process intensifications for CO_2_ absorption processes have been investigated by employing selective membrane materials [14] and using MEA as an absorbent. The membrane absorption efficiency depending on the distribution coefficient was carried out with the properties of absorbents [15], and thus, was obtained by combining both chemical absorption and separation techniques simultaneously according to the diffusion-reaction model [13]. The one-dimensional steady-state modeling equation was successfully applied to predict the CO_2_ absorption flux under various operational conditions associated with occurring reactions by using amines as absorbents [16]. Moreover, an effective strategy was investigated to capture CO_2_ in turbulent flow conditions [17] in the one-dimensional steady-state modeling equation [16,18] as compared to considering the laminar flow velocity of the liquid profile. The implementation of membrane contactors in the gas absorption process aims to allow the soluble gas mixture components to be selectively absorbed in hollow-fiber gas/liquid membrane contactors [19,20] by the solvent on the membrane surface of the liquid phase [11,13]. Experimental studies [21,22] on shell-side mass transfer performance in hollow-fiber membrane modules were reviewed recently by Lipnizki and Field [23]. The fiber spacing, fiber diameters, and inlet and outlet effects were examined, and the device performance varied significantly. The fiber distribution and flow distribution in randomly packed fiber bundles were investigated [24,25], and the analytical solution and the experimental runs for shell side mass transfer with fluid flowing axially between fiber cells were studied by Zheng et al. [26].

A two-dimensional mathematical statement was developed theoretically and experimentally in a hollow-fiber membrane gas/liquid absorption module [27], with the gas and liquid flow rates regulated independently. It is believed that the availability of such a simplified mathematical formulation as developed here for hollow-fiber membrane absorption module is the value in the present work and will be a significant contribution to design and model multi-stream separation devices associating mutual conditions at the boundary. The chemical absorption of CO_2_ by MEA was carried out and illustrated to validate the theoretical predictions, and theoretical treatment on the assumption based on the shell side mass transfer of an ordered fiber arrangement was developed in the present study. The resultant partial differential equations, referred to as conjugated Graetz problems [28,29], were solved analytically using the separated variable method associated with an orthogonal expansion technique [30,31]. The absorption efficiency was evaluated using the absorbent flow rate (Graetz number) and the inlet CO_2_ concentration in the gas feed as parameters. The theoretical predictions show that the effect of the inlet CO_2_ concentration in the gas feed plays an important role in absorption efficiency [32], and the absorption efficiency improvement of the hollow-fiber membrane contactor is obtained by implementing more fiber cells under various absorbent flow rates. The influences of operating and design parameters on absorption flux and absorption efficiency are also delineated.

The primary aims in this study are to develop the two-dimensional mathematical formulation of a hollow-fiber membrane contactor in an MEA absorbent system and to obtain the solutions analytically to be validated with experimental data under both concurrent- and countercurrent-flow operations. The theoretical predictions are presented graphically with the packing density (φ), absorbent Graetz number (volumetric flow rate), and flow pattern as parameters. The effects of the MEA absorbent flow rate and inlet CO_2_ concentration on the averaged Sherwood number, absorption flux, and absorption efficiency were investigated theoretically and experimentally.

## 2. Theoretical Formulation

Happel’s free surface appeared in the imaginary outer boundary of the cell [26,33]. A fiber cell model with an imaginary free surface was developed to describe the mass transport between the shell side with one fiber in each cell of the hollow fiber module. The model building was assumed to be uniformly packed, with the bundle’s porosity equal to the fluid’s envelope porosity, no friction on the shell side, and ignoring the velocity profile across the module radius direction. The outside radius of the fiber cell and free surface are $r_{0}$ and $r_{f}$, respectively, as shown in Figure 1 for being simplified into a circular-tube module.

The module was assumed to be regularly packed, and the velocity profile in the radial direction was ignored. Comparing with the radius $r_{f}$ of imaginary free surface, the thickness of the hydrophobic microporous membrane δ is negligible (δ << $r_{f}$). The overall mass transfer regions, including (a) CO_2_ gas transfers into the membrane surface from the fiber cell; (b) CO_2_ diffuses through the membrane pores; (c) CO_2_ transfers into the liquid stream via the membrane/liquid interface; (d) CO_2_ reacted by MEA liquid solutions. The mathematical formulations of the transport phenomena of the laminar mass transfer problem for this small fiber cell belong to the conjugated Graetz problem category, which was derived at a steady state with negligible axial diffusion. The velocity distributions and conservation equations were formulated according to the following assumptions:(1)Steady state and fully developed flow in each flowing channel;(2)Negligible axial diffusion and conduction, entrance length, and end effects;(3)Happel’s surface model used to characterize the velocity profile in the fiber cell;(4)Isothermal operation and constant physical properties;(5)The applicability of thermodynamic equilibrium and Henry’s law;(6)The chemical reaction is very fast and the equilibrium state is reached;(7)The hollow fiber membrane thickness can be neglected as compared to the hollow fiber radius.

### 2.1. Concurrent-Flow Operations

The hollow fiber module can be reduced to a circular-tube module according to Happel’s free surface model [26,33], as indicated in Figure 2 of the hollow-fiber membrane contactor. The four regions considered for modeling CO_2_ absorption in hollow-fiber membrane contactors are shown in Figure 3. The convective velocity in the radial direction is neglected, and the axial velocity distributions are derived by applying the continuity equation and Navier-Stokes relations to obtain the hydro-dynamical equation for laminar flow. Thus, the dimensionless equations derived by the conservation equations of mass with specified velocities may be written in terms of the dimensionless variables as
(1)$$v_{a}\left(\eta_{a}\right)=2\bar{v_{a}}\left[1-{\left(\frac{\eta_{a}}{\eta_{i}}\right)}^{2}\right]$$
(2)$$v_{b}\left(\eta_{b}\right)=\frac{2\bar{v_{b}}}{\left[\left(\frac{2}{\eta_{m}^{2}}-3\right)+\eta_{o}^{2}\right]}\left[\eta_{o}^{2}-{\left(1-\eta_{b}\right)}^{2}+2ln\left(\frac{1-\eta_{b}}{\eta_{o}}\right)\right]$$
and
(3)$$\left[\frac{v_{a}r_{f}^{2}}{LD_{a}}\right]\frac{\partial \psi_{a}\left(\eta_{a},\xi \right)}{\partial \xi }=\frac{1}{\eta_{a}}\left[\frac{\partial }{\partial \eta_{a}}\left(\eta_{a}\frac{\partial \psi_{a}\left(\eta_{a},\xi \right)}{\partial \eta_{a}}\right)\right]$$
(4)$$\left[\frac{v_{b}r_{f}^{2}}{LD_{b}}\right]\frac{\partial \psi_{b}\left(\eta_{b},\xi \right)}{\partial \xi }=\frac{1}{\left(1-\eta_{b}\right)}\left[\frac{\partial }{\partial \eta_{b}}\left(\left(1-\eta_{b}\right)\frac{\partial \psi_{b}\left(\eta_{b},\xi \right)}{\partial \eta_{b}}\right)\right]-k_{CO_{2}}\psi_{b}\left(\eta_{b},\xi \right)$$
in which $r_{o}$ is the fiber outside radius, $r_{f\;}$ is the free surface radius, L channel length, and φ is the packing density of the hollow fiber module with the following dimensionless groups
(5)ηa=rarf, ηb=rbrf, ηi=rirf, ηo=rorf, ξ=zL, rf=φ−0.5ro,    ψa=CaCai−Cbi, ψb=CbCai−Cbi, Gza=va¯rf2LDa, Gzb=vb¯rf2LDb, ηm=1−ηo22ln(1ηo)

The boundary conditions required for solving Equations (3) and (4) are
(6)$$\psi_{a}\left(\eta_{a},0\right)=\psi_{ai}$$
(7)$$\psi_{b}\left(\eta_{b},0\right)=\psi_{bi}$$
(8)$$\frac{\partial \psi_{a}\left(0,\xi \right)}{\partial \eta_{a}}=0$$
(9)$$\frac{\partial \psi_{b}\left(0,\xi \right)}{\partial \eta_{b}}=0$$
(10)$$-\frac{\partial \psi_{a}\left(\eta_{i},\xi \right)}{\partial \eta_{a}}=\frac{\varepsilon \;r_{f}}{\delta }\left[\psi_{a}\left(\eta_{i},\xi \right)-\frac{K_{ex}^{\prime }}{H}\psi_{b}\left(1-\eta_{o},\xi \right)\right]$$
(11)$$-\frac{\partial \psi_{a}\left(\eta_{i},\xi \right)}{\partial \eta_{a}}=\frac{\eta_{o}D_{b}}{\eta_{i}D_{a}}\frac{\partial \psi_{b}\left(1-\eta_{o},\xi \right)}{\partial \eta_{b}}$$
where $D_{a}$ is ordinary diffusion coefficient of CO_2_ in N_2_, $D_{b}$ is ordinary diffusion coefficient of CO_2_ in MEA, ε is the porosity of membrane, and *H =* 0.73 is dimensionless Henry’s law constant [8]. The reduced equilibrium constant $K_{ex}^{’}$ is derived to fit in the modeling equation as
(12)$$K_{ex}^{’}=K_{ex}\left[\mathrm{MEA}]/[H^{+}\right]$$
in which the equilibrium constant $K_{ex}=\left[{\mathrm{MEACOO}}^{-}\right]\left[H^{+}\right]/\left\{\left[{\mathrm{CO}}_{2}\right]\left[\mathrm{MEA}\right]\right\}=1.25\times 10^{-5}$ at T=298 K [34] in the CO_2_ absorption from gas phase by aqueous MEA absorbent, and can be expressed as follows:(13)$${\mathrm{CO}}_{2}+\mathrm{MEA}↔{\mathrm{MEACOO}\;}^{-}+H^{+}$$

The present work is actually the extension of our previous work [35] by following the similar general solution form except instead of the hollow-fiber membrane contactors, but the mathematical formulation is more complicated with using the Happel’s free surface model than that in the parallel-plate module. The analytical solutions of dimensionless concentration distributions of both phases, $\psi_{a}$ and $\psi_{b}$, may be obtained by the use of an orthogonal expansion technique with the eigenfunction expanding in terms of an extended power series. By following the same mathematical procedure performed in previous works [30,36], the variables are separated in the form:(14)$$\psi_{a}\left(\eta_{a},\xi \right)=\sum_{m=0}^{\infty }S_{a,m}F_{a,m}\left(\eta_{a}\right)G_{m}\left(\xi \right)$$
(15)$$\psi_{b}\left(\eta_{b},\xi \right)=\sum_{m=0}^{\infty }S_{b,m}F_{b,m}\left(\eta_{b}\right)G_{m}\left(\xi \right)$$

Substitution of Equations (14) and (15) into Equations (3) and (4) gives
(16)$$G_{m}\left(\xi \right)=e^{\lambda_{m}\xi }$$
(17)$$F_{a,m}^{\prime \prime }\left(\eta_{a}\right)+\frac{1}{\eta_{a}}F_{a,m}^{’}\left(\eta_{a}\right)-\left[\frac{v_{a}\left(\eta_{a}\right)r_{f}^{2}}{LD_{a}}\right]\lambda_{m}F_{a,m}\left(\eta_{a}\right)=0$$
(18)$$F_{b,m}^{\prime \prime }\left(\eta_{b}\right)-\frac{1}{\left(1-\eta_{b}\right)}F_{b,m}^{’}\left(\eta_{b}\right)-\left[\frac{v_{b}\left(\eta_{b}\right)r_{f}^{2}}{LD_{AC}}\right]\lambda_{m}F_{b,m}\left(\eta_{b}\right)-k_{CO_{2}}F_{b,m}\left(\eta_{b}\right)=0$$
and the boundary conditions in Equations (6)–(9) can be rewritten as
(19)$$F_{a,m}^{’}\left(0\right)=0$$
(20)$$F_{b,m}^{’}\left(0\right)=0$$
(21)$$-S_{a,m}F_{a,m}^{’}\left(\eta_{i}\right)=\frac{\varepsilon \;r_{f}}{\delta }\left[S_{a,m}F_{a,m}\left(\eta_{i}\right)-\frac{K_{ex}^{’}}{H}S_{b,m}F_{b,m}\left(1-\eta_{o}\right)\right]$$
(22)$$-S_{a,m}F_{a,m}^{’}\left(\eta_{i}\right)=\frac{\eta_{o}D_{b}}{\eta_{i}D_{a}}S_{b,m}F_{b,m}^{’}\left(1-\eta_{o}\right)$$
where the primes on $F_{a,m}\left(\eta_{a}\right)$ and $F_{b,m}\left(\eta_{b}\right)$ denote the differentiations with respect to $\eta_{a}$ and $\eta_{b}$, respectively. The eigen-functions $F_{a,m}\left(\eta_{a}\right)$ and $F_{b,m}\left(\eta_{b}\right)$ were assumed to be polynomials to avoid the loss of generality as follows:(23)$$F_{a,m}\left(\eta_{a}\right)=\sum_{n=0}^{\infty }d_{mn}\eta_{a}^{n},\;d_{m0}=1\;(\mathrm{selected}),\;d_{m1}=0$$
(24)$$F_{b,m}\left(\eta_{b}\right)=\sum_{n=0}^{\infty }e_{mn}\eta_{b}^{n},\;e_{m0}=1\;(\mathrm{selected}),\;e_{m1}=0$$

Equation (21) can be rearranged to acquire the relationship between expansion coefficients $S_{a,m}$ and $S_{b,m}$ as
(25)$$S_{b,m}=\frac{H\left[\delta \;F_{a,m}^{’}\left(\eta_{i}\right)+\varepsilon \;r_{f}F_{a,m}\left(\eta_{i}\right)\right]}{\varepsilon \;r_{f}K_{ex}^{’}F_{b,m}\left(1-\eta_{o}\right)}S_{a,m}$$

Moreover, rearranging and deleting $S_{a,m}$ and $S_{b,m}$ in Equations (21) and (22) to calculate the eigenvalue $\lambda_{m}$ ($\lambda_{1}$,$\lambda_{2}$,…., $\lambda_{m}$,….) in the following equation
(26)$$-F_{a,m}^{’}\left(\eta_{i}\right)=\frac{\varepsilon \;r_{f}}{\delta }\left[F_{a,m}\left(\eta_{i}\right)+\frac{\eta_{i}K_{ex}^{’}D_{a}}{H\eta_{o}D_{b}}\frac{F_{a,m}^{’}\left(\eta_{i}\right)F_{b,m}\left(1-\eta_{o}\right)}{F_{b,m}^{’}\left(1-\eta_{o}\right)}\right]$$

Substituting Equations (23) and (24) into Equations (21) and (22), all the coefficients $d_{mn}$ and $e_{m,n}$ may be expressed in terms of eigenvalues $\lambda_{m}$ after using Equations (8) and (9). The term $ln\left[\left(1-\eta_{b}\right)/\eta_{o}\right]$ in velocity distributions of Equation (2) can be approximated using the six-term Taylor series for acceptable tolerance as follows:(27)$$ln\left(1-\eta_{b}\right)=-\eta_{b}-\frac{\eta_{b}^{2}}{2}-\frac{\eta_{b}^{3}}{3}-\frac{\eta_{b}^{4}}{4}-\frac{\eta_{b}^{5}}{5}-\frac{\eta_{b}^{6}}{6}$$

Combining Equations (17)–(20), (23)–(24), and (27), all the coefficients $d_{mn}$ and $e_{m,n}$ can be expressed in terms of eigenvalue $\lambda_{m}$ as
(28)$$d_{m2}=\frac{Gz_{a}\lambda_{m}}{2},\;d_{m\;3}=0,\;d_{mn}=\frac{2Gz_{a}\lambda_{m}}{n\left(n-1\right)}\left[d_{m\left(n-2\right)}-\frac{1}{\eta_{i}^{2}}d_{m\left(n-4\right)}\right],\;n=4,5,6,\ldots$$
and
(29)$$e_{m,2}=\frac{1}{2}\left(k_{CO_{2}}+STGz_{b}\lambda_{m}\right),\;e_{m,3}=\frac{1}{6}\left(k_{CO_{2}}+STGz_{b}\lambda_{m}\right),\;\;\;\;\;\;\;\;e_{m,n}=\frac{n-1}{n}e_{m,n-1}+\frac{SGz_{b}\lambda_{m}}{n\left(n-1\right)}(Te_{m,n-2}-Te_{m,n-3}+2e_{m,n-4}+\frac{3}{4}e_{m,n-5}\;\;+\frac{1}{6}e_{m,n-6}+\frac{1}{10}e_{m,n-7}+\frac{1}{15}e_{m,n-8}+\frac{1}{3}e_{m,n-9})+k_{CO_{2}}\left(e_{m,n-2}-e_{m,n-3}\right),n=4,5,6,\ldots$$
in which $S=2/\left[\left(\frac{2}{\eta_{m}^{2}}-3\right)+\eta_{o}^{2}\right]$, $T=\eta_{o}^{2}-2ln\eta_{o}-1$.

These eigenvalues $\lambda_{m}$ were calculated in Equation (26), resulting in both positive and negative sets under concurrent and countercurrent-flow operations. Table 1 shows that the calculation results of the first five eigenvalues and their associated expansion coefficients are illustrated to meet the convergence requirement within the acceptable truncation error with the series terms *n* = 400 for $Q_{a}=3.33\times 10^{-6}\;m^{3}/s$ and $Q_{b}=10.0\times 10^{-6}\;m^{3}/s$. The eigen-functions associated with the corresponding eigenvalues are also well defined by Equations (23) and (24), once all eigenvalues were obtained from Equation (26). These eigenvalues $\lambda_{m}$ include a negative set, which is required for both concurrent- and countercurrent-flow operations; the eigenvalues indicated in Table 1 are the dominant set in the system.

The mathematical treatment is similar to that in the previous works [30,36]. The orthogonality condition in the double-flow gas-liquid membrane contactor system of the case with $\lambda_{m}\neq \lambda_{n}$ is verified as follows:(30)$$\eta_{o}D_{a}\int_{0}^{\eta_{i}}\left[\frac{v_{a}r_{f}{}^{2}}{LD_{a}}\right]S_{a,i}S_{a,i}\eta_{a}F_{a,j}F_{a,j}d\eta_{a}+\frac{\eta_{i}D_{b}}{H}\int_{0}^{1-\eta_{o}}\left[\frac{v_{b}r_{f}^{2}}{LD_{b}}\right]S_{b,i}S_{b,j}\left(1-\eta_{b}\right)F_{b,i}F_{b,j}d\eta_{b}=0$$

The dimensionless inlet and outlet stream concentrations expanded to the sum of an infinite series according to Equations (14) and (15) with the use of boundary conditions as
(31)$$\psi_{a}\left(\eta ,0\right)=\sum_{m=0}^{\infty }S_{a,m}F_{a,m}\left(\eta \right)=\psi_{ai}$$
(32)$$\psi_{b}\left(\eta ,0\right)=\sum_{m=0}^{\infty }S_{b,m}F_{b,m}\left(\eta \right)=\psi_{bi}$$

Multiplying both sides of Equations (31) and (32) at ξ=0 by $S_{a,n}F_{a,n}\eta_{o}\eta_{a}D_{a}\left(v_{a}r_{f}^{2}/LD_{a}\right)$ and $S_{a,n}F_{a,n}\left[\eta_{i}\left(1-\eta_{b}\right)D_{b}K_{ex}^{’}/H\right]\left(v_{b}r_{f}^{2}/LD_{b}\right)$, respectively, and integrating summing together to obtain the general expression for the expansion coefficients in the following relationship accordingly
(33)$$\eta_{o}D_{a}\int_{0}^{\eta_{i}}\eta_{a}\left[\frac{v_{a}r_{f}^{2}}{LD_{a}}\right]S_{a,n}F_{a,n}\psi_{ai}d\eta_{a}+\frac{\eta_{i}D_{b}K_{ex}^{’}}{H}\int_{0}^{1-\eta_{o}}\left(1-\eta_{b}\right)\left[\frac{v_{b}r_{f}^{2}}{LD_{b}}\right]S_{b,n}F_{b,n}\psi_{bi}d\eta_{b}\;\;\;\;\;\;\;\;\;\;\;=\eta_{o}D_{a}\int_{0}^{\eta_{i}}\eta_{a}\left[\frac{v_{a}r_{f}^{2}}{LD_{a}}\right]S_{a,n}F_{a,n}\sum_{m=0}^{\infty }S_{a,m}F_{a,m}d\eta_{a}+\frac{\eta_{i}D_{b}K_{ex}^{’}}{H}\int_{0}^{1-\eta_{o}}\left(1-\eta_{b}\right)\left[\frac{v_{b}r_{f}^{2}}{LD_{b}}\right]S_{b,n}F_{b,n}\sum_{m=0}^{\infty }S_{b,m}F_{b,m}d\eta_{b}$$

The summation terms in Equation (33) were dropped out due to using the orthogonality condition
(34)$$\eta_{o}D_{a}\int_{0}^{\eta_{i}}\eta_{a}\left[\frac{v_{a}r_{f}^{2}}{LD_{a}}\right]S_{a,n}F_{a,n}\psi_{ai}d\eta_{a}+\frac{\eta_{i}D_{b}K_{ex}^{’}}{H}\int_{0}^{1-\eta_{o}}\left(1-\eta_{b}\right)\left[\frac{v_{b}r_{f}^{2}}{LD_{b}}\right]S_{b,n}F_{b,n}\psi_{bi}d\eta_{b}\;=\eta_{o}D_{a}\int_{0}^{\eta_{i}}\eta_{a}\left[\frac{v_{a}r_{f}^{2}}{LD_{a}}\right]S_{a,n}^{2}F_{a,n}^{2}d\eta_{a}+\frac{\eta_{i}D_{b}K_{ex}^{’}}{H}\int_{0}^{1-\eta_{o}}\left(1-\eta_{b}\right)\left[\frac{v_{b}r_{f}^{2}}{LD_{b}}\right]S_{b,n}^{2}F_{b,n}^{2}d\eta_{b}$$

Substitution of Equation (25) to replace $S_{b,n}$ into Equation (34) and dividing $S_{a,n}\eta_{o}D_{a}$ result in
(35)$$\int_{0}^{\eta_{i}}\eta_{a}\left[\frac{v_{a}r_{f}^{2}}{LD_{a}}\right]F_{a,n}\psi_{ai}d\eta_{a}+\frac{\eta_{i}D_{b}\left[\delta \;F_{a,n}^{’}\left(\eta_{i}\right)+\varepsilon \;r_{f}F_{a,n}\left(\eta_{i}\right)\right]}{\eta_{o}D_{a}\varepsilon \;r_{f}F_{b,n}\left(1-\eta_{o}\right)}\int_{0}^{1-\eta_{o}}\left(1-\eta_{b}\right)\left[\frac{v_{b}r_{f}^{2}}{LD_{a}}\right]F_{b,n}\psi_{bi}d\eta_{b}\;\;\;\;\;\;=S_{a,n}\left\{\int_{0}^{\eta_{i}}\eta_{a}\left[\frac{v_{a}r_{f}^{2}}{LD_{a}}\right]F_{a,n}^{2}d\eta_{a}+\frac{\eta_{i}D_{b}H{\left[\delta \;F_{a,n}^{’}\left(\eta_{i}\right)+\varepsilon \;r_{f}F_{a,n}\left(\eta_{i}\right)\right]}^{2}}{\eta_{o}D_{a}K_{ex}^{’}\varepsilon^{2}r_{f}^{2}F_{b,n}^{2}\left(1-\eta_{o}\right)}\int_{0}^{1-\eta_{o}}\left(1-\eta_{b}\right)\left[\frac{v_{b}r_{f}^{2}}{LD_{b}}\right]F_{b,n}^{2}d\eta_{b}\right\}$$

The expansion coefficient was thus obtained at ξ=0 as follows:(36)$$S_{a,n}=\frac{\int_{0}^{\eta_{i}}\eta_{a}\left[\frac{v_{a}r_{f}^{2}}{LD_{a}}\right]F_{a,n}\psi_{ai}d\eta_{a}+\frac{\eta_{i}D_{b}\left[\delta \;F_{a,n}^{’}\left(\eta_{i}\right)+\varepsilon \;r_{f}F_{a,n}\left(\eta_{i}\right)\right]}{\eta_{o}D_{b}\varepsilon \;r_{f}F_{b,n}\left(1-\eta_{o}\right)}\int_{0}^{1-\eta_{o}}\left(1-\eta_{b}\right)\left[\frac{v_{b}r_{f}^{2}}{LD_{b}}\right]F_{b,n}\psi_{bi}d\eta_{b}}{\int_{0}^{\eta_{i}}\eta_{a}\left[\frac{v_{a}r_{f}^{2}}{LD_{a}}\right]F_{a,n}^{2}d\eta_{a}+\frac{\eta_{i}D_{b}H{\left[\delta \;F_{a,n}^{’}\left(\eta_{i}\right)+\varepsilon \;r_{f}F_{a,n}\left(\eta_{i}\right)\right]}^{2}}{\eta_{o}D_{a}K_{ex}^{’}\varepsilon^{2}r_{f}^{2}F_{b,n}^{2}\left(1-\eta_{o}\right)}\int_{0}^{1-\eta_{o}}\left(1-\eta_{b}\right)\left[\frac{v_{b}r_{f}^{2}}{LD_{b}}\right]F_{b,n}^{2}d\eta_{b}}$$

Similarly, the boundary conditions at ξ=1 were expressed in terms of infinite series with the use of Equations (14) and (15)
(37)$$\psi_{a}\left(\eta_{a},1\right)=\sum_{m=0}^{\infty }S_{a,m}F_{a,m}e^{\lambda_{m}}$$
(38)$$\psi_{b}\left(\eta_{b},1\right)=\sum_{m=0}^{\infty }S_{b,m}F_{b,m}e^{\lambda_{m}}$$

Manipulating both sides of Equations (37) and (38) at ξ=1 and performing the same procedure at the boundary condition at ξ=0 gives the following relationship of the general expression for the expansion coefficients as
(39)$$\int_{0}^{\eta_{i}}\eta_{a}\left[\frac{v_{a}r_{f}^{2}}{LD_{a}}\right]F_{a,n}\psi_{a}\left(\eta_{a},1\right)d\eta_{a}+\frac{\eta_{i}D_{b}\left[\delta \;F_{a,n}^{’}\left(\eta_{i}\right)+\varepsilon \;r_{f}F_{a,n}\left(\eta_{i}\right)\right]}{\eta_{o}D_{a}\varepsilon \;r_{f}F_{b,n}\left(1-\eta_{o}\right)}\int_{0}^{1-\eta_{o}}\left(1-\eta_{b}\right)\left[\frac{v_{b}r_{f}^{2}}{LD_{b}}\right]F_{b,n}\psi_{b}\left(\eta_{b},1\right)d\eta_{b}\;=S_{a,n}\left\{\int_{0}^{\eta_{i}}\eta_{a}\left[\frac{v_{a}r_{f}^{2}}{LD_{a}}\right]F_{a,n}^{2}d\eta_{a}+\frac{\eta_{i}D_{b}H{\left[\delta \;F_{a,n}^{’}\left(\eta_{i}\right)+\varepsilon \;r_{f}F_{a,n}\left(\eta_{i}\right)\right]}^{2}}{\eta_{o}D_{a}K_{ex}^{’}\varepsilon^{2}r_{f}^{2}F_{b,n}^{2}\left(1-\eta_{o}\right)}\int_{0}^{1-\eta_{o}}\left(1-\eta_{b}\right)\left[\frac{v_{b}r_{f}^{2}}{LD_{b}}\right]F_{b,n}^{2}d\eta_{b}\right\}$$
or the expansion coefficient at ξ=1 was given by
(40)$$S_{a,n}=e^{-\lambda_{n}}\frac{\int_{0}^{\eta_{i}}\eta_{a}\left[\frac{v_{a}r_{f}^{2}}{LD_{a}}\right]F_{a,n}\psi_{a}\left(\eta_{a},1\right)d\eta_{a}+\frac{\eta_{i}D_{b}\left[\delta \;F_{a,n}^{’}\left(\eta_{i}\right)+\varepsilon \;r_{f}F_{a,n}\left(\eta_{i}\right)\right]}{\eta_{o}D_{a}\varepsilon \;r_{f}F_{b,n}\left(1-\eta_{o}\right)}\int_{0}^{1-\eta_{o}}\left(1-\eta_{b}\right)\left[\frac{v_{b}r_{f}^{2}}{LD_{b}}\right]F_{b,n}\psi_{b}\left(\eta_{b},1\right)d\eta_{b}}{\int_{0}^{\eta_{i}}\eta_{a}\left[\frac{v_{a}r_{f}^{2}}{LD_{a}}\right]F_{a,n}^{2}d\eta_{a}+\frac{\eta_{i}D_{b}H{\left[\delta \;F_{a,n}^{’}\left(\eta_{i}\right)+\varepsilon \;r_{f}F_{a,n}\left(\eta_{i}\right)\right]}^{2}}{\eta_{o}D_{a}\varepsilon^{2}r_{f}^{2}F_{b,n}^{2}\left(1-\eta_{o}\right)}\int_{0}^{1-\eta_{o}}\left(1-\eta_{b}\right)\left[\frac{v_{b}r_{f}^{2}}{LD_{b}}\right]F_{b,n}^{2}d\eta_{b}}$$

Both numerators of the expansion coefficients $S_{a,n}$ are equal, and equating Equations (36) and (40) at both inlet and outlet of the feed stream in the gas-liquid membrane contactor to give
(41)$$\int_{0}^{\eta_{i}}\eta_{a}\left[\frac{v_{a}r_{f}^{2}}{LD_{a}}\right]F_{a,n}\psi_{ai}d\eta_{a}+\frac{\eta_{i}D_{b}\left[\delta \;F_{a,n}^{’}\left(\eta_{i}\right)+\varepsilon \;r_{f}F_{a,n}\left(\eta_{i}\right)\right]}{\eta_{o}D_{a}\varepsilon \;r_{f}F_{b,n}\left(1-\eta_{o}\right)}\int_{0}^{1-\eta_{o}}\left(1-\eta_{b}\right)\left[\frac{v_{b}r_{f}^{2}}{LD_{b}}\right]F_{b,n}\psi_{bi}d\eta_{b}\;\;\;\;\;\;\;\;\;=e^{-\lambda_{n}}\left\{\int_{0}^{\eta_{i}}\eta_{a}\left[\frac{v_{a}r_{f}^{2}}{LD_{a}}\right]F_{a,n}^{2}d\eta_{a}+\frac{\eta_{i}D_{b}H{\left[\delta \;F_{a,n}^{’}\left(\eta_{i}\right)+\varepsilon \;r_{f}F_{a,n}\left(\eta_{i}\right)\right]}^{2}}{\eta_{o}D_{a}K_{ex}^{’}\varepsilon^{2}r_{f}^{2}F_{b,n}^{2}\left(1-\eta_{o}\right)}\int_{0}^{1-\eta_{o}}\left(1-\eta_{b}\right)\left[\frac{v_{b}r_{f}^{2}}{LD_{b}}\right]F_{b,n}^{2}d\eta_{b}\right\}$$

Now, the outlet concentrations on the right-hand side of Equation (41) may be expressed according to Equations (14) and (15) for concurrent-flow operations as follows:(42)$$\psi_{a}\left(\eta ,1\right)=\sum_{q=0}^{\infty }S_{a,q}F_{a,q}e^{-\lambda_{q}}$$
(43)$$\psi_{b}\left(\eta ,1\right)=\sum_{q=0}^{\infty }S_{b,q}F_{b,q}e^{-\lambda_{q}}$$

Substitutions of Equations (42) and (43) into Equation (41) with the use of Equation (25) give
(44)$$\psi_{ai}\int_{0}^{\eta_{i}}\eta_{a}\left[\frac{v_{a}r_{f}^{2}}{LD_{a}}\right]F_{a,n}d\eta_{a}+\frac{\eta_{i}D_{b}\left[\delta \;F_{a,n}^{’}\left(\eta_{i}\right)+\varepsilon \;r_{f}F_{a,n}\left(\eta_{i}\right)\right]\psi_{bi}}{\eta_{o}D_{a}\varepsilon \;r_{f}F_{b,n}\left(1-\eta_{o}\right)}\int_{0}^{1-\eta_{o}}\left(1-\eta_{b}\right)\left[\frac{v_{b}r_{f}^{2}}{LD_{b}}\right]F_{b,n}d\eta_{b}\;\;\;\;\;\;\;\;\;\;\;=\sum_{q=0}^{\infty }S_{a,q}e^{\lambda_{q}-\lambda_{n}}\left\{\int_{0}^{\eta_{i}}\eta_{a}\left[\frac{v_{a}r_{f}^{2}}{LD_{a}}\right]F_{a,n}F_{a,q}d\eta_{a}\right.\;\;\;\;\;\;\;\;\;\;\;\;\;\;\;\;\;\;\;+\frac{\eta_{i}D_{b}H\left[\delta F_{a,n}^{’}\left(\eta_{i}\right)+\varepsilon r_{f}F_{a,n}\left(\eta_{i}\right)\right]\;\left[\delta F_{a,q}^{’}\left(\eta_{i}\right)+\varepsilon r_{f}F_{a,q}\left(\eta_{i}\right)\right]}{\eta_{o}D_{a}K_{ex}^{’}\varepsilon^{2}r_{f}^{2}F_{b,n}\left(1-\eta_{o}\right)F_{b,q}\left(1-\eta_{o}\right)}\left.\int_{0}^{1-\eta_{o}}\left[\frac{v_{b}r_{f}^{2}}{LD_{b}}\right]\left(1-\eta_{b}\right)F_{b,n}F_{b,q}d\eta_{b}\right\}$$

The expansion coefficients $S_{b,q}$ are obtained by following the same derivation procedure [37] with integrating Equation (44) with the aid of Equations (14) and (15) once $S_{a,n}$ was calculated as shown in Equation (25) by acquiring the relationship between $S_{a,n}$ and $S_{b,n}$ as follows:

(I)When n=0
(45)$$\frac{Gz_{a}\eta_{i}^{2}\psi_{ai}}{2}+\frac{Gz_{b}\eta_{i}D_{a}\psi_{bi}}{2\eta_{o}D_{b}}=S_{a,0}\left(\frac{Gz_{a}\eta_{i}^{2}}{2}+\frac{Gz_{b}\eta_{i}D_{a}H}{2\eta_{o}D_{b}K_{ex}^{’}}\right)\;\;\;\;\;\;\;\;\;\;\;\;\;\;\;\;+\sum_{q=1}^{\infty }S_{a,q}\frac{\eta_{i}e^{\lambda_{q}}}{\lambda_{q}}\left\{F_{a,q}^{’}\left(\eta_{i}\right)+\frac{D_{a}H\left[\delta F_{a,q}^{’}\left(\eta_{i}\right)+\varepsilon r_{f}F_{a,q}\left(\eta_{i}\right)\right]}{D_{b}K_{ex}^{’}\varepsilon r_{f}}\frac{F_{b,q}^{’}\left(1-\eta_{o}\right)}{F_{b,q}\left(1-\eta_{o}\right)}\right\}$$(II)When n≠0, n=q
(46)$$\frac{\psi_{ai}}{\lambda_{n}}\eta_{i}F_{a,n}^{’}\left(\eta_{i}\right)+\frac{\psi_{bi}\eta_{i}D_{b}\left[\delta F_{a,n}^{’}\left(\eta_{i}\right)+\varepsilon r_{f}F_{a,n}\left(\eta_{i}\right)\right]F_{b,n}^{’}\left(1-\eta_{o}\right)}{\lambda_{n}D_{a}\varepsilon r_{f}F_{b,n}\left(1-\eta_{o}\right)}\;\;\;\;\;\;\;\;\;\;\;=\sum_{q=1}^{\infty }S_{a,q}\eta_{i}\left\{\left[\frac{\partial F_{a,n}^{’}}{\partial \lambda_{n}}\left(\eta_{i}\right)F_{a,n}\left(\eta_{i}\right)-\frac{\partial F_{a,n}}{\partial \lambda_{n}}\left(\eta_{i}\right)F_{a,n}^{’}\left(\eta_{i}\right)\right]\right.\;\;\;\;\;\;\;\;\;\;\;\;\;\;\;\;\;\;\;\;+\frac{D_{b}H{\left[\delta F_{a,n}^{’}\left(\eta_{i}\right)+\varepsilon r_{f}F_{a,n}\left(\eta_{i}\right)\right]}^{2}}{D_{a}K_{ex}^{’}\varepsilon^{2}r_{f}^{2}F_{b,n}^{2}\left(1-\eta_{o}\right)}\left.\left[\frac{\partial F_{b,n}^{’}}{\partial \lambda_{n}}\left(1-\eta_{o}\right)F_{b,n}\left(1-\eta_{o}\right)-\frac{\partial F_{b,n}}{\partial \lambda_{n}}\left(1-\eta_{o}\right)F_{b,n}^{’}\left(1-\eta_{o}\right)\right]\right\}$$(III)When n≠0, n≠q
(47)$$\frac{\psi_{ai}}{\lambda_{n}}\eta_{i}F_{a,n}^{’}\left(\eta_{i}\right)+\frac{\psi_{bi}\eta_{i}D_{b}\left[\delta F_{a,n}^{’}\left(\eta_{i}\right)+\varepsilon r_{f}F_{a,n}\left(\eta_{i}\right)\right]F_{b,n}^{’}\left(1-\eta_{o}\right)}{\lambda_{n}D_{a}\varepsilon r_{f}F_{b,n}\left(1-\eta_{o}\right)}\;\;\;\;\;\;\;\;\;\;\;=S_{a,0}\frac{\eta_{i}e^{-\lambda_{n}}}{\lambda_{n}}\left\{F_{a,n}^{’}\left(\eta_{i}\right)+\frac{D_{b}H\left[\delta F_{a,n}^{’}\left(\eta_{i}\right)+\varepsilon r_{f}F_{a,n}\left(\eta_{i}\right)\right]}{D_{a}K_{ex}^{’}\varepsilon r_{f}}\frac{F_{b,n}^{’}\left(1-\eta_{o}\right)}{F_{b,n}\left(1-\eta_{o}\right)}\right\}\;\;\;\;\;+\sum_{q=1}^{\infty }S_{a,q}\frac{\eta_{i}e^{\lambda_{q}-\lambda_{n}}}{\lambda_{n}-\lambda_{q}}\left\{\left[F_{a,n}^{’}\left(\eta_{i}\right)F_{a,q}\left(\eta_{i}\right)-F_{a,n}\left(\eta_{i}\right)F_{a,q}^{’}\left(\eta_{i}\right)\right]\right.\;\;\;\;\;\;\;\;\;\;\;\;\;\;\;\;+\frac{D_{b}H\left[\delta F_{a,n}^{’}\left(\eta_{i}\right)+\varepsilon r_{f}F_{a,n}\left(\eta_{i}\right)\right]\;\left[\delta F_{a,q}^{’}\left(\eta_{i}\right)+\varepsilon r_{f}F_{a,q}\left(\eta_{i}\right)\right]}{D_{a}K_{ex}^{’}\varepsilon^{2}r_{f}^{2}}\left.\left[\frac{F_{b,n}^{’}\left(1-\eta_{o}\right)}{F_{b,n}\left(1-\eta_{o}\right)}-\frac{F_{b,q}^{’}\left(1-\eta_{o}\right)}{F_{b,q}\left(1-\eta_{o}\right)}\right]\right\}$$

### 2.2. Countercurrent-Flow Operations

The governing equations of mass transfer for describing countercurrent-flow operations are exactly the same as that in concurrent-flow operations of Equations (3) and (4), except for the velocity distribution of Equation (2) and the boundary condition of Equation (7) being replaced by
(48)$$v_{b}\left(\eta_{b}\right)=\frac{-2\bar{v_{b}}}{\left[\left(\frac{2}{\eta_{m}^{2}}-3\right)+\eta_{o}^{2}\right]}\left[\eta_{o}^{2}-{\left(1-\eta_{b}\right)}^{2}+2ln\left(\frac{1-\eta_{b}}{\eta_{o}}\right)\right]$$
(49)$$\psi_{b}\left(\eta_{b},1\right)=\psi_{bi}$$

By following the same derivation performed in the previous section of concurrent-flow operations, the results of the expansion coefficient of $S_{a,n}$ and $S_{b,n}$ can be obtained as follows:

(I)When n=0
(50)$$\frac{Gz_{a}\eta_{i}^{2}\psi_{ai}}{2}+\frac{Gz_{b}\eta_{i}D_{a}\psi_{bi}}{2\eta_{o}D_{b}}=S_{a,0}\left[\frac{Gz_{a}\eta_{i}^{2}}{2}+\frac{Gz_{b}\eta_{i}D_{a}H}{2\eta_{o}D_{b}K_{ex}^{’}}\right]\;\;\;\;\;\;\;\;\;\;\;\;\;\;\;\;+\sum_{q=1}^{\infty }S_{a,q}\frac{\eta_{i}}{\lambda_{q}}\left\{F_{a,q}^{’}\left(\eta_{i}\right)-\frac{e^{\lambda_{q}}D_{a}H\left[\delta F_{a,q}^{’}\left(\eta_{i}\right)+\varepsilon r_{f}F_{a,q}\left(\eta_{i}\right)\right]}{D_{b}K_{ex}^{’}\varepsilon r_{f}F_{b,q}\left(1-\eta_{o}\right)}F_{b,q}^{’}\left(1-\eta_{o}\right)\right\}$$(II)When n≠0 and n=q
(51)$$\frac{e^{-\lambda_{n}}\psi_{ai}}{\lambda_{n}}\eta_{i}F_{a,n}^{’}\left(\eta_{i}\right)-\frac{\psi_{bi}\eta_{i}D_{b}\left[\delta F_{a,n}^{’}\left(\eta_{i}\right)+\varepsilon r_{f}F_{a,n}\left(\eta_{i}\right)\right]F_{b,n}^{’}\left(1-\eta_{o}\right)}{\lambda_{n}D_{a}\varepsilon r_{f}F_{b,n}\left(1-\eta_{o}\right)}\;\;\;\;\;\;\;\;\;\;\;\;\;\;\;\;\;=\sum_{q=1}^{\infty }S_{a,q}\eta_{i}\left\{\left[\frac{\partial F_{a,n}^{’}}{\partial \lambda_{n}}\left(\eta_{i}\right)F_{a,n}\left(\eta_{i}\right)-\frac{\partial F_{a,n}}{\partial \lambda_{n}}\left(\eta_{i}\right)F_{a,n}^{’}\left(\eta_{i}\right)\right]\right.-\frac{D_{b}H{\left[\delta F_{a,n}^{’}\left(\eta_{i}\right)+\varepsilon r_{f}F_{a,n}\left(\eta_{i}\right)\right]}^{2}}{D_{a}K_{ex}^{’}\varepsilon^{2}r_{f}^{2}F_{b,n}^{2}\left(1-\eta_{o}\right)}\;\left.\times \left[\frac{\partial F_{b,n}^{’}}{\partial \lambda_{n}}\left(1-\eta_{o}\right)F_{b,n}\left(1-\eta_{o}\right)-\frac{\partial F_{b,n}}{\partial \lambda_{n}}\left(1-\eta_{o}\right)F_{b,n}^{’}\left(1-\eta_{o}\right)\right]\right\}\;\;\;\;\;\;\;\;$$(III)When n≠0 and n≠q
(52)$$\frac{e^{-\lambda_{n}}\psi_{ai}}{\lambda_{n}}\eta_{i}F_{a,n}^{’}\left(\eta_{i}\right)-\frac{\psi_{bi}\eta_{i}D_{b}\left[\delta F_{a,n}^{’}\left(\eta_{i}\right)+\varepsilon r_{f}F_{a,n}\left(\eta_{i}\right)\right]F_{b,n}^{’}\left(1-\eta_{o}\right)}{\lambda_{n}D_{a}\varepsilon r_{f}F_{b,n}\left(1-\eta_{o}\right)}\;\;\;\;\;\;\;\;\;\;\;\;\;\;=S_{a,0}\frac{\eta_{i}}{\lambda_{n}}\left\{F_{a,n}^{’}\left(\eta_{i}\right)-\frac{e^{-\lambda_{n}}D_{b}H\left[\delta F_{a,n}^{’}\left(\eta_{i}\right)+\varepsilon r_{f}F_{a,n}\left(\eta_{i}\right)\right]}{D_{a}K_{ex}^{’}\varepsilon r_{f}F_{b,n}\left(1-\eta_{o}\right)}F_{b,n}^{’}\left(1-\eta_{o}\right)\right\}\;\;\;\;\;\;\;\;\;\;\;+\sum_{q=1}^{\infty }S_{a,q}\frac{\eta_{i}}{\lambda_{n}-\lambda_{q}}\left\{\left[F_{a,n}^{’}\left(\eta_{i}\right)F_{a,q}\left(\eta_{i}\right)-F_{a,n}\left(\eta_{i}\right)F_{a,q}^{’}\left(\eta_{i}\right)\right]\right.\;\;\;\;\;\;\;\;\;\;\;\;\;\;\;\;-\frac{e^{\lambda_{q}-\lambda_{n}}D_{b}H\left[\delta F_{a,n}^{’}\left(\eta_{i}\right)+\varepsilon r_{f}F_{a,n}\left(\eta_{i}\right)\right]\;\left[\delta F_{a,q}^{’}\left(\eta_{i}\right)+\varepsilon r_{f}F_{a,q}\left(\eta_{i}\right)\right]}{D_{a}K_{ex}^{’}\varepsilon^{2}r_{f}^{2}}\left.\left[\frac{F_{b,n}^{’}\left(1-\eta_{o}\right)}{F_{b,n}\left(1-\eta_{o}\right)}-\frac{F_{b,q}^{’}\left(1-\eta_{o}\right)}{F_{b,q}\left(1-\eta_{o}\right)}\right]\right\}$$

### 2.3. Absorbent Efficiency in the Gas/Liquid Membrane Absorption System

The local Sherwood number in the absorbent stream is defined by
(53)$$Sh_{b\xi }=\frac{k_{b\xi }D_{eq,b}}{D_{b}}$$
in which and $D_{eq,b}=2\left(r_{f}-r_{o}\right)$ is the equivalent diameter of the shell side and the local mass transfer coefficient $k_{b\xi }$ of gas stream is defined by
(54)$$k_{b\xi }=\frac{D_{b}}{r_{f}}\frac{\partial \psi_{b}\left(1-\eta_{o},\xi \right)/\partial \eta_{b}}{\psi_{b}\left(1-\eta_{o},\xi \right)-\bar{\psi_{b}}\left(\xi \right)}$$

The final expression of the local Sherwood number is obtained in Equation (55)
(55)$$Sh_{b\xi }=\frac{k_{b\xi }D_{eq,b}}{D_{b}}=\frac{2\left(1-\eta_{o}\right)\sum_{m=1}^{\infty }S_{b,m}F_{b,m}^{’}\left(1-\eta_{o}\right)e^{\lambda_{m}\xi }}{\sum_{m=1}^{\infty }S_{b,m}\left[F_{b,m}\left(1-\eta_{o}\right)-\frac{2\eta_{o}}{Gz_{b}\lambda_{m}}F_{b,m}^{’}\left(1-\eta_{o}\right)\right]e^{\lambda_{m}\xi }}$$

Therefore, the average Sherwood number can be obtained as
(56)$$\bar{Sh_{b}}=\int_{0}^{1}Sh_{b\xi }d\xi =\int_{0}^{1}\frac{2\left(1-\eta_{o}\right)\sum_{m=1}^{\infty }S_{b,m}F_{b,m}^{’}\left(1-\eta_{o}\right)e^{\lambda_{m}\xi }}{\sum_{m=1}^{\infty }S_{b,m}\left[F_{b,m}\left(1-\eta_{o}\right)-\frac{2\eta_{o}}{Gz_{b}\lambda_{m}}F_{b,m}^{’}\left(1-\eta_{o}\right)\right]e^{\lambda_{m}\xi }}d\xi$$

The absorption flux *J* and absorption efficiency $I_{M}$ are defined by the total amount of the CO_2_ transferred from the fiber cell to the shell side per unit area, and the percentage of the initial CO_2_ left in the initial gas phase, respectively, which can be determined using Equations (57) and (58) as follows
(57)$$J=Q_{a}\left(\bar{C_{ai}}-\bar{C_{ae}}\right)/N\left(2\pi r_{o}L_{b}\right)$$
(58)$$I_{M}=\frac{\bar{C_{ai}}-\bar{C_{ae}}}{\bar{C_{ai}}}\times 100\%$$

## 3. Membrane Modularization and Experimental Setup

The experimental results were monitored to validate the theoretical predictions derived from the mathematical models derived in previous section. A photo of a more detailed configuration of the concentric-tube membrane contactor module is presented in Figure 4. A gas mixture containing CO_2_ and N_2_ was introduced from the well gas mixing tank, where N_2_ and CO_2_ feed in through the tube side, while 30 wt% MEA ($5.0\times 10^{-3}{\;\mathrm{mol}/\;\mathrm{cm}}^{3}$) solution was chosen and regulated by a flow meter (MB15GH-4-1, Fong-Jei, New Taipei, Taiwan) between 5.0~10.0 cm^3^/s (5.0, 5.67, 8.33, 10.0 cm^3^/s) to supply the liquid absorbent flowing into the shell side from the reservoir. The positive pressure difference of the MEA solution side was kept higher than that of the CO_2_/N_2_ gas mixture side to avoid bubbling. The CO_2_/N_2_ gas feed flow rates introduced from the gas mixing tank (EW-06065-02, Cole Parmer Company, Vernon Hills, IL, USA) and regulated by using the mass flow controller (N12031501PC-540, Protec, Brooks Instrument, Hatfield, PA, USA) were controlled at 3.33 cm^3^/s with three inlet CO_2_ concentrations of 30%, 35%, and 40%, respectively. The CO_2_ concentration in the inlet and outlet streams was collected and measured for comparisons using gas chromatography (Model HY 3000 Chromatograp, China Corporation, New Taipei, Taiwan). The experimental apparatus of the CO_2_ absorption using MEA absorbent flowing into the hydrophobic alumina hollow-fiber membrane modules with a porosity of ε = 0.55, a thickness of δ = 250 μm, and a nominal pore size of 0.2 μm as illustrated in Figure 5. Figure 5a,b illustrates schematic representations of the hollow-fiber gas-liquid membrane contactor systems for concurrent- and countercurrent-flow operations, respectively, in which the MEA solution passes through the shell side and the gas feed flows into the tube side.

The parameters that include the volumetric flow rate of both the gas feed and liquid absorbent ($Q_{a\;}$ and $Q_{b\;}$), membrane contactor module ($r_{s\;}$, $r_{i\;}$, $r_{o\;}$, *L* and *N*), permeability of membrane (ε), solute diffusivity both in gas feed and liquid absorbent ($D_{a\;}$ and $D_{b\;}$), and Henry’s law constant ($H_{\;}$) were provided in this study. The inner radius of the module shell is of $r_{s}$ = 0.0075 m, and the inner and outer radius of the fiber cell are $r_{i}$ = 0.0004 m and $r_{o}$ = 0.00065 m, respectively. The inorganic hydrophobic membrane was used in the experiments $r_{o}$ = 0.00065 m for its superior chemical resistance and thermal stability. The alumina hollow fiber membranes were prepared in a combined dry-wet spinning and phase inversion method followed by a sintering process. The hollow fiber precursors were fabricated by spinning alumina slurry comprised of alumina powders (0.7 µm, α-Al_2_O_3_, Alfa Aesar, 99.9% metal basis), N-Methyl-2-pyrrolidone (NMP, TEDIA, Echo Chemical, Taiwan, purity > 99%), polyethersulfone (PES, Veradel A-301, SOLVAY, Trump Chemical, Taiwan, amber color), and polyethyleneglycol 30-dipolyhydroxystearate (Arlacel P135, Croda Taiwan, Taiwan, molecular weight: 5000 g mol^−1^), which are used as the main ceramic materials, solvent, binder, and dispersant, respectively. The Al_2_O_3_:NMP:PES:P135 molar ratio in the slurry was 5:4:1:0.1. Briefly, the P135 paste was first added to the NMP solution and vigorously stirred until a homogenous solution was formed. Next, the alumina powder was gradually added to the solution and stirred well. Subsequently, PES pellets were added to the solution and stirred until the PES was completely dissolved. Finally, a homogenous spinning slurry was obtained. In our spinning process, deionized water (DI) was used as a non-solvent for phase inversion purposes.

The as-prepared slurry and DI water were coextruded through a tube-in-orifice spinneret with an inner diameter of 0.7 mm and an outer diameter of 2.0 mm. The orifice side and tube side were for slurry with a flow rate of 15 mL/min and DI water with a flow rate of 10 mL/min, respectively. The nascent fiber passed through an air gap of 10 cm and went into a coagulation bath of DI water for external solvent exchange. The air gap was to allow the phase inversion to occur first from the inner surface of the nascent fiber. Rapid precipitation occurs at the inner fiber surface, resulting in long fingers. Usually, the opening pores of finger-like structures are larger than the voids (pores for ceramic membranes) of particle packing. Those larger pores were not favored in this study. Thus, by introducing the air gap, most solvents were exchanged, and the solidification of the slurry phase was almost done before entering the coagulation bath. This could greatly reduce the formation of finger-like structures of fibers starting from the outer surface. A coagulation bath was used to make sure all solvents were exchanged and the precipitation of the polymer of the slurry was completed. The membrane precursors were obtained after 2-day water immersion for completed polymer precipitation and were then debinded at 480 °C for 12 h (ramp rate of 1.6 °C min^−1^) and sintered at 1400 °C for 2 h (ramp rate of 2 °C min^−1^) to form porous alumina hollow fiber membranes in a shell-and-tube type glass module. The as-prepared porous alumina hollow fiber membranes were first cut to 0.17 m in length. The alumina hollow fiber membrane sets with different packing densities can be obtained by encapsulating different numbers of fibers. Three membrane sets with implementing 7 fiber cells and 19 fiber cells, respectively, were fabricated in this work. The fiber cells were fixed with a particular arrangement in the module by sealing both ends of the tube side using thermoset epoxy. The pinch clamps and tubing for the membrane module are shown in Figure 6.

## 4. Results and Discussion

### 4.1. Outlet Concentration Distributions

Following the mathematical treatment performed in the previous works [26,36], the procedure for calculating the theoretical values of the dimensionless outlet average concentration, absorption rate, and absorption efficiency are described as follows. First, for concurrent-flow operations, the eigenvalues $\lambda_{m}$ ($\lambda_{1}$,$\lambda_{2}$,…., $\lambda_{m}$,….) in the membrane contactor are solved from Equation (26), the associated eigen-functions the associated eigenfunctions ($F_{a,m}\left(\eta_{a}\right)$ and $F_{b,m}\left(\eta_{b}\right)$, m=0, 1, 2,….) obtained from Equations (23) and (24) with the coefficients in Equations (28) and (29). Next, combined with the expansion coefficients ($S_{a,m}$ and $S_{b,m}$, m=0, 1, 2,….), as shown in Equations (40) and (45)–(47), the concentration distribution of gas feed or liquid absorbent, $\psi_{a}\left(\eta_{a},\xi \right)$ and $\psi_{b}\left(\eta_{b},\xi \right)$, are readily obtained in Equations (14) and (15). Lastly, the radially averaged concentrations for absorbent and gas streams of both concurrent- and countercurrent-flow operations are calculated from Equations (53) and (54), while the absorption rate and absorption efficiency are calculated from Equations (44), (57), and (58), respectively. Some results for using the 19 fiber cell module under countercurrent-flow operations as well as the dimensionless outlet concentration are shown in Table 1. Only the first three eigenvalues, as well as their corresponding eigenfunctions, are necessary to be included during the calculation procedure due to rapid convergence, as indicated in Table 1. Figure 7a,b shows the dimensionless averaged outlet CO_2_ concentration $\bar{\psi_{ae}}$ profiles for various inlet CO_2_ concentrations and the mass-transfer Graetz number $Gz_{b}$ of MEA absorbent under both concurrent- and countercurrent-flow operations with implementing *N* = 7 fiber cells as an illustration. Note that the dimensionless averaged outlet concentration distribution increased with the inlet CO_2_ concentration. The comparison reveals that the higher the absorbent Graetz number $Gz_{b}$ of MEA absorbent, the lower the averaged outlet CO_2_ concentration found in both calculation and measurement, as predicted. The results show that a higher driving-force concentration gradient is kept between two phases under a larger inlet CO_2_ concentration, leading to a higher absorption rate for both flow patterns. One can find that the dimensionless average outlet concentration of the countercurrent-flow operations is lower than that of concurrent-flow operations. Thus, the descending absorption flux along the flowing channel for the concurrent-flow operations is thus confirmed compared to a higher total absorption rate in the countercurrent-flow operations.

Figure 8 presents the dependence of the Sherwood number $\bar{Sh_{b}}$ on the absorbent Graetz number $Gz_{b}$. The averaged Sherwood number $\bar{Sh_{b}}$ plays a significant role in determining the CO_2_ absorption rate when considering mass transfer behavior. The theoretical average Sherwood numbers with the absorbent Graetz number $Gz_{b}$ in MEA absorbent as a parameter, as shown in Figure 8.

The theoretical prediction $\bar{Sh_{b}}$ increases with an increase in the absorbent Graetz number for both concurrent- and countercurrent-flow operations, as presented in Figure 8. The results show that the averaged Sherwood number $\bar{Sh_{b}}$ in countercurrent-flow operations is higher than that in concurrent-flow operations. This result also confirms that the higher mass-transfer coefficient is obtained in countercurrent-flow operations that come up with a lower outlet CO_2_ concentration $\bar{\psi_{ae}}$. The value of the averaged Sherwood number $\bar{Sh_{b}}$ in the countercurrent-flow configurations with a larger significant concentration gradient is higher than that in the concurrent-flow configurations due to utilizing the driving-force concentration gradient more effectively. Notably, the implementing fibers significantly increased the averaged Sherwood number $\bar{Sh_{b}}$ and the absorption rate for both modules with implementing *N* = 7 fiber cells and *N* = 19 fiber cells, respectively. Despite the effect on the number of fiber cells, the change in the flow patterns only led to a moderate effect on the change in the Sherwood number.

### 4.2. Absorption Flux and Absorption Efficiency

Theoretical predictions for the CO_2_ absorption flux versus the MEA absorbent Graetz number with inlet CO_2_ concentration and flow pattern as parameters under $Q_{a}=3.33{\;\mathrm{cm}}^{3}/s$, as indicated in Figure 9. The experimental results of the absorption flux shown in Figure 9 prove the validity by defining the accuracy deviation E of the theoretical predictions from the experimental results are within an acceptable range, as indicated in Table 2, with the definition as follows:(59)$$E\left(\%\right)=\frac{1}{N_{exp}}\sum_{i=1}^{N_{exp}}\frac{\left|\overset{\wedge }{J_{i}}-J_{i}\right|}{\overset{\wedge }{J_{i}}}$$
where $\overset{\wedge }{J_{i}}$ indicates the theoretical prediction of J while $N_{exp}$ and $J_{i}$ are the number of the experimental measurements and the experimental data of J, respectively. The accuracy derivations between the experimental results and theoretical predictions for concurrent- and countercurrent-flow operations in Figure 9 are $4.10\times 10^{-2}\leq E\leq 1.50\times 10^{-2}$ and $1.40\times 10^{-2}\leq E\leq 9.0\times 10^{-1}$, respectively, as presented in Table 2. Both qualitative and quantitative agreements were achieved between the theoretical predictions and the experimental results of this study.

The increase of the absorbent Graetz number creates the positive effect not only enhancing the absorption flux but also reducing the outlet CO_2_ concentration in the fiber cells, as indicated from Figure 9. This absorption flux may be attributed to the increasing MEA absorbent Graetz number and thus the convective mass-transfer coefficient to suppress the disadvantage effect of concentration polarization on the membrane surface. A lower mass-transfer resistance and a larger driving-force concentration gradient between both sides of the membrane surfaces were achieved.

Comparisons were made on theoretical predictions of absorption efficiencies between the hollow fiber modules with implementing *N* = 7 fiber cells and *N* = 19 fiber cells under both concurrent- and countercurrent-flow operations, respectively, as shown in Figure 10. The results show that the absorption efficiency increases with the increase of the MEA absorbent Graetz number but decreases with the inlet CO_2_ concentration, and the extent of the absorption efficiency increment is more significant in countercurrent-flow operations. Notice that the effect of the number of fiber cells on absorption efficiency concludes that there is a larger absorption efficiency with implementing more fiber cells into hollow fiber modules. The absorption efficiency of gas/liquid contactor is improving when $Gz_{b}$ is increasing, as confirmed in Figure 10 as well as in the reported gas absorption processes. The present work extends the existing study, except for using hollow-fiber membrane contactors instead of parallel-plate membrane contactors [36] under the same inlet CO_2_ concentrations (30%, 35%, and 40%). The comparison of absorption efficiencies in both modules indicated that the present design of using hollow-fiber membrane contactors is preferred. Overall, the performance of the hollow fiber membrane absorption module is enhanced by implementing fiber cells into the shell tube. In other words, inserting more fiber cells into the shell tube gives a higher value of absorption efficiency, which reflects that a more effective device performance in increasing the total absorption rate is expected. Although the absorption flux of the operations with a 40% inlet CO_2_ concentration in Figure 9 is higher than that of a 30% inlet CO_2_ concentration, the absorption efficiency with an inlet CO_2_ concentration is in reverse order. The results also indicate that the absorption efficiency in the countercurrent-flow configuration with a more significant concentration gradient is higher than in the concurrent-flow configuration.

## 5. Conclusions

A hollow-fiber membrane module for CO_2_ absorption that implemented various numbers of fiber cells to enhance the absorption flux was investigated theoretically and experimentally. The theoretical predictions of concentration distributions of CO_2_ absorption were developed in the form of mathematical formulations by making mass balances of both gas feed stream and MEA absorbent with the absorbent Greatz number as a parameter. This study further examines device performance by evaluating the absorption flux J and absorption efficiency $I_{M}$ of the hollow fiber module by implementing the fiber cells into the MEA absorbent stream, which reached a significant achievement under both concurrent- and countercurrent-flow operations, as demonstrated in Figure 9 and Figure 10, respectively. The theoretical predictions of the averaged Sherwood number and absorption efficiency accomplished in the present study were predicted analytically without the aid of experimental runs, as shown in Equations (56) and (58), respectively. The comparisons of the absorption efficiency were drawn to the following conclusions:●The absorption increases with the increase of the MEA absorbent Graetz number.●The absorption efficiency is obtained by implementing fiber cells where the absorption rate enhancement of *N* = 19 fiber cells is higher than that of *N* = 7 fiber cells but increases with decreasing the inlet CO_2_ concentration.●The absorption flux increases with an increase in the number of fiber cells and the inlet CO_2_ concentration.●A more considerable absorption flux is achieved in countercurrent-flow operations than that in concurrent-flow operations due to utilizing the driving-force concentration gradient more effectively.●Fore eigenvalues were used in the calculation procedure, and a good approximation was obtained, as indicated in Table 1. The results show that the agreement is fairly good in predicting the theoretical predictions, with an accuracy of $4.10\times 10^{-2}\leq E\leq 9.0\times 10^{-1}$ for the absorption flux.

It is worth noting that this theoretical modeling may also be applied to other hollow fiber modules in membrane separation processes that have not previously been studied and verified by experimental results.

## Figures and Tables

**Figure 1 membranes-12-01021-f001:**
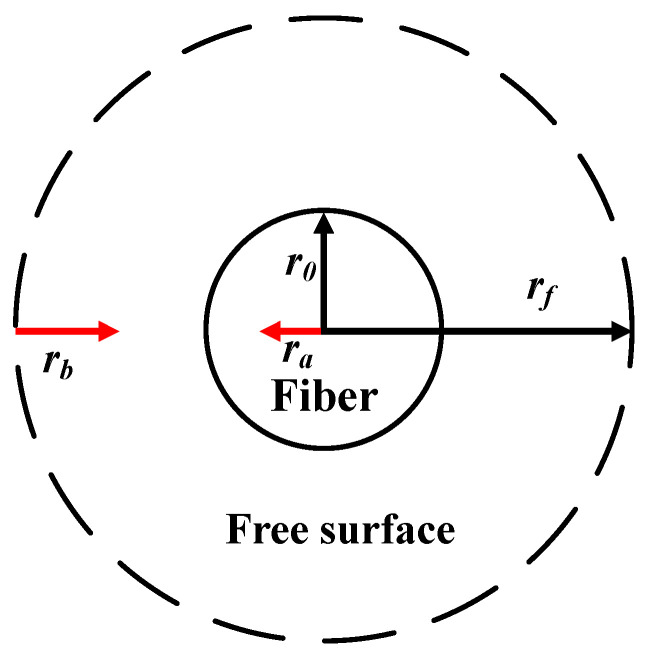
A scheme for the free surface model.

**Figure 2 membranes-12-01021-f002:**
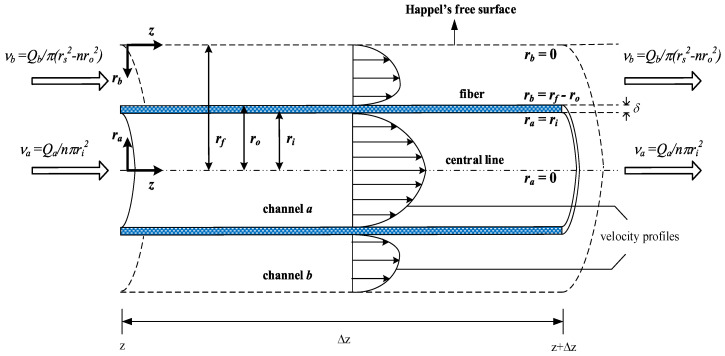
Schematic diagram of Happel’s free surface in a hollow-fiber gas-liquid membrane module.

**Figure 3 membranes-12-01021-f003:**
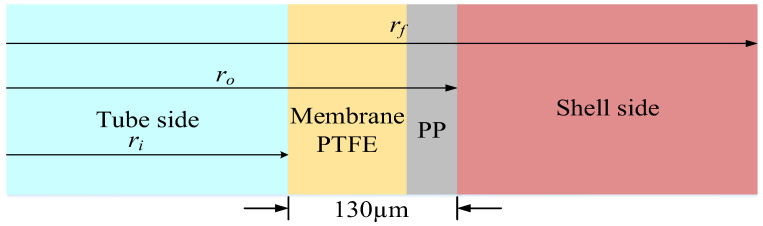
Four regions considered for modeling CO_2_ absorption in hollow-fiber membrane contactors.

**Figure 4 membranes-12-01021-f004:**
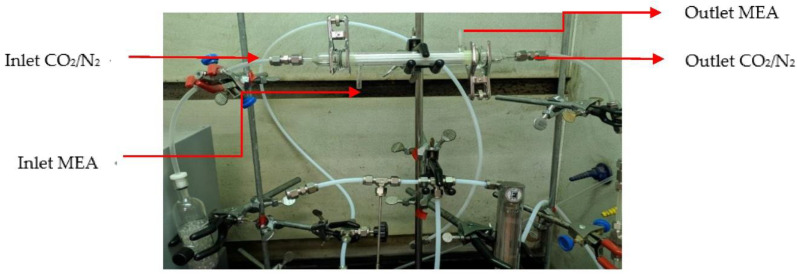
Photo of the experimental setup for the CO_2_ absorption membrane module.

**Figure 5 membranes-12-01021-f005:**
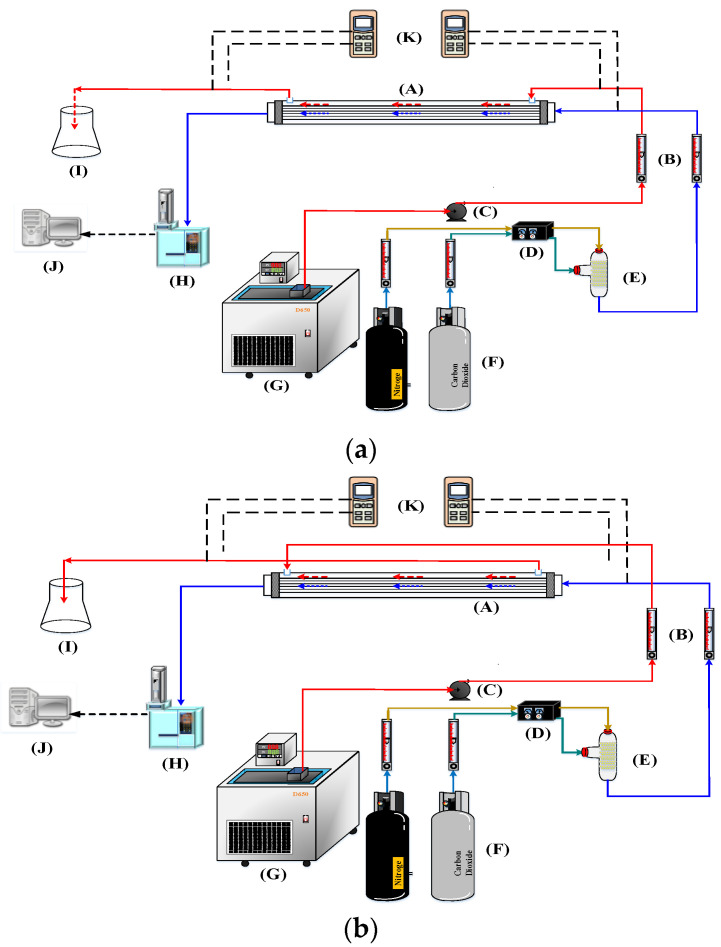
Schematic diagram of the experimental setup for CO_2_ absorption with MEA by hollow-fiber gas–liquid membrane contactors. (**a**) Concurrent-flow operation; (**b**) Countercurrent-flow operation. (A) Hollow fiber membrane module; (B) Flow meter; (C) Pump; (D) Mass flow controller; (E) Mixer; (F) Gas cylinder; (G) Thermostatic tank; (H) Chromatograph; (I) Beaker; (J) Monitor; (K) Temperature indicator.

**Figure 6 membranes-12-01021-f006:**
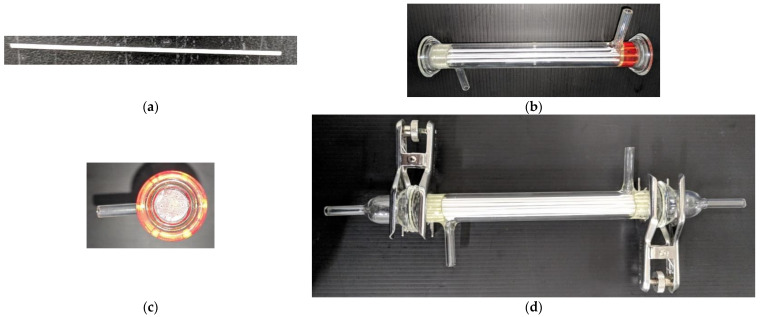
Details of the configuration of the hollow fiber membrane module. (**a**) Fiber cell; (**b**) A bunch of fiber cells in a circular tube; (**c**) Wrapped cap; (**d**) Circular hollow-fiber module.

**Figure 7 membranes-12-01021-f007:**
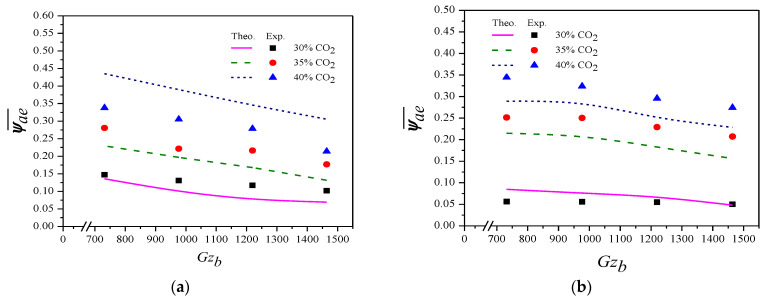
Effects of $Gz_{b}$ and inlet CO_2_ concentration on outlet CO_2_ concentration ($Gz_{a}=710$). (**a**) Concurrent-flow operations; (**b**) Countercurrent-flow operations.

**Figure 8 membranes-12-01021-f008:**
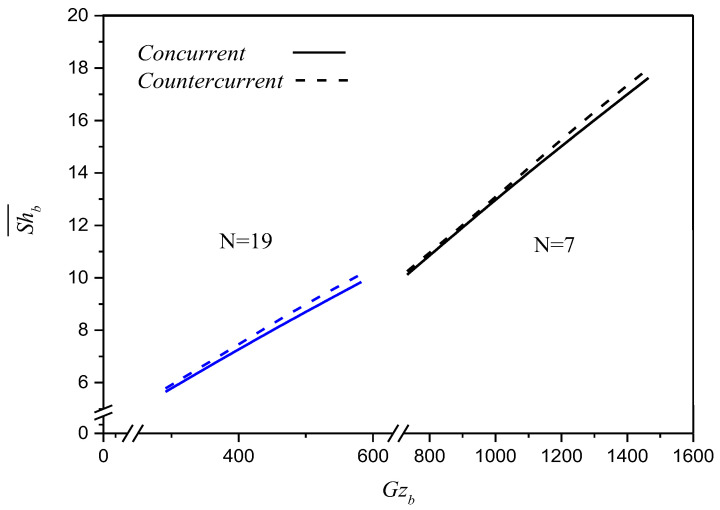
Effects of fiber number, flow pattern and $Gz_{b}$ on theoretical averaged Sherwood number.

**Figure 9 membranes-12-01021-f009:**
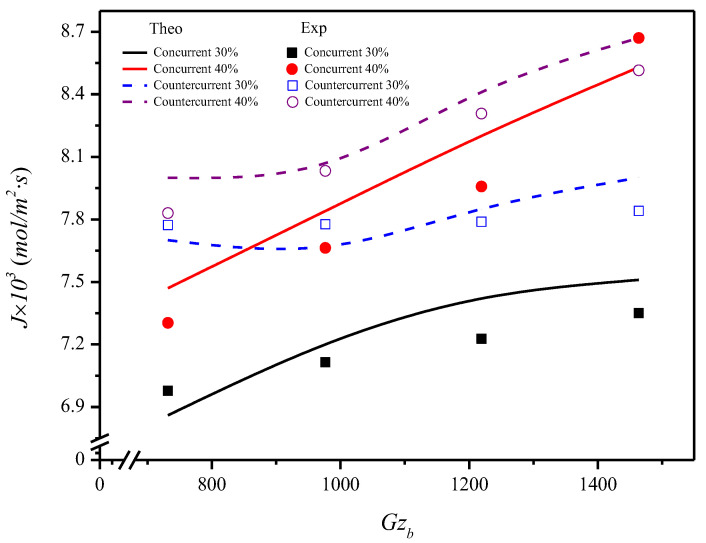
Effects of inlet CO_2_ concentration, flow pattern and $Gz_{b}$ on CO_2_ absorption flux.

**Figure 10 membranes-12-01021-f010:**
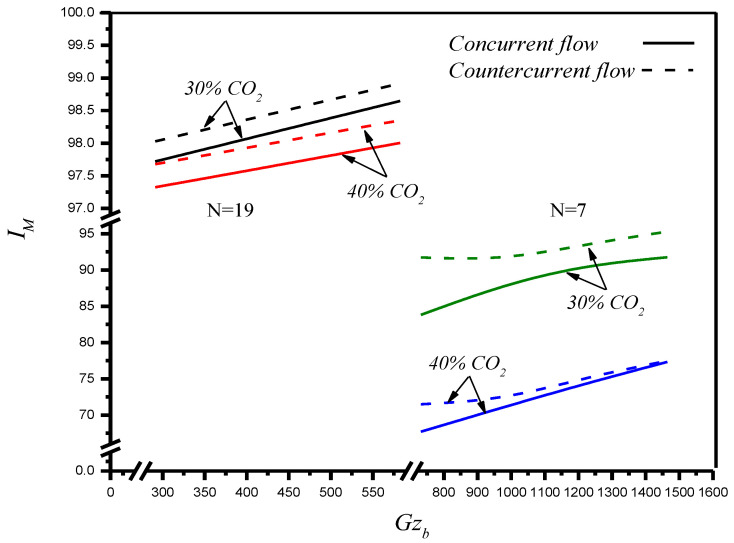
Effects of fiber number, inlet CO_2_ concentration, flow pattern and $Gz_{b}$ on CO_2_ absorption flux (*N* = 7 and *N* = 19).

**Table 1 membranes-12-01021-t001:** The dimensionless outlet concentrations and the associated eigenvalues and expansion coefficients under countercurrent-flow operations with 19 fiber cells.

m	$\lambda_{0}$	$\lambda_{1}$	$\lambda_{2}$	$\lambda_{3}$	$\lambda_{4}$	$\lambda_{5}$	$S_{a,0}$	$S_{a,1}$	$S_{a,2}$	$S_{a,3}\times 10^{3}$	$S_{a,4}\times 10^{4}$	$S_{a,5}\times 10^{5}$	$\bar{\psi_{ae}}$
*n* = 300													
3	0.0	−0.199	−4.113	−13.814	-	-	0.063	0.021	−0.080	7.06	-	-	0.1719
4	0.0	−0.199	−4.113	−13.814	−29.453	-	0.063	0.021	−0.080	7.06	−0.296	-	0.1620
5	0.0	−0.199	−4.113	−13.814	−29.453	−51.063	0.064	0.021	−0.080	7.06	−0.296	−0.927	0.1620
*n* = 400													
3	0.0	−0.132	−3.857	−13.455	-	-	0.457	0.106	−0.082	9.98	-	-	0.0181
4	0.0	−0.132	−3.857	−13.455	10.206	-	0.457	0.106	−0.082	9.98	−3.11	-	0.0199
5	0.0	−0.132	−3.857	−13.455	10.206	29.023	0.457	0.106	−0.082	9.98	−3.11	−4.20	0.0199

**Table 2 membranes-12-01021-t002:** The accuracy of the experimental results.

CO_2_ (%)	$E\;\left(\%\right)$			
	*N* = 7		*N* = 19	
	Concurrent	Countercurrent	Concurrent	Countercurrent
30	1.90	1.30	4.10	0.90
35	2.30	1.00	1.50	1.10
40	1.40	1.40	1.50	0.90

## Data Availability

Not applicable.

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
