# Peer review of "Two-Dimensional Conjugated Mass Transfer of Carbon Dioxide Absorption in a Hollow-Fiber Gas-Liquid Membrane Contactor"

_membranes, 2022, doi:10.3390/membranes12101021_

Round 1
Reviewer 1 Report
The authors did a great job at thoroughly and clearly describing their methodology and results in this manuscript. The manuscript is well-written and very professional. Their work will be an excellent contribution to further progress membrane contactors for carbon capture. I recommend the following revisions to further strengthen and clarify the paper:
1. Please update the introduction with more recent references. Currently, all the works cited are from 2010 or earlier.
2. Update the introduction to stress the novelty and contribution of this work. It is unclear what knowledge gaps this manuscript is addressing by studying the chemical absorption of CO2 by MEA using hollow fiber membrane contactors. The current state of the introduction makes it seems like this work is very similar to previous works, even stating so throughout the text (for example, lines 160 and 210).
3. All the figures need to provide more information in the captions. For example, Figure 2 should state the notation is for a cross-flow direction and that it is defining the velocity profiles within a single fiber. Another example is Figure 4, where the reader would benefit from the use of labels or text when looking at the experimental setup.
4. Figure 3 does not provide additional information/support and can be removed.
5. When discussing assumptions, please include the Happel’s free surface model assumption that allows the approach to be applied to this application: assumes the bundle’s porosity is equal to the fluid’s envelope porosity and assumes no friction on the shell-side.
6. It would help the reader follow the thorough theoretical formulation if the variables are defined within the text.
7. Figure 6 seems randomly placed in the experimental section. Please clarify how the figure facilitates this section or move it to the computational section.
8. Expand the captions for all the result plots, i.e., describing what the lines/symbols/colors signify and providing an overall take-away message for each plot.
9. I recommend using a bullet list in the conclusions section to help organize the final message.
Author Response
Dear Professor Tung,
Attached file is the revised manuscript entitled, “membranes-1967801, Two-Dimensional Conjugated Mass Transfer of Carbon Dioxide Absorption in a Hollow-Fiber Gas-Liquid Membrane Contactor”, which has been submitted to Membranes (Special Issue: Selected Papers from the 16th International Conference on Inorganic Membranes (ICIM)). It has been carefully revised and edited in which reviewers’ comments and your suggestions are incorporated. All the revisions in the revised manuscript are marked with font color in red. An Itemized Response to the Reviewers’ Comments is also enclosed.
I would like to express our thanks for the valuable comments and inputs from you and this scientific community. We hope that this revised manuscript would meet the Journal’s expectation and be considered for publication on Membranes. Your further consideration is greatly appreciated.
Best regards,
Chii-Dong Ho, Ph.D.
Distinguished Professor, Department of Chemical and Materials Engineering
Tamkang University
Tamsui, Taipei
Taiwan 251

Reviewer 2 Report
This work studied the absorption efficiency of CO2 in hollow fiber membrane contactor using MEA solvent under counter current and parallel current operation, and also carried out theoretical and experimental research. The operation of hollow fiber membrane contactor also provides a cheap way to improve the absorption efficiency by increasing the number of fibers and considering the equipment performance. This work is interesting but needs to be modified slightly. Suggestions are as follows:
1、Why choose amine solution as chemical absorbent? What are the advantages compared with other chemical absorbents?
2、What is the purpose of external solvent exchange for new fibers entering the coagulation bath of deionized water through a 10 cm air gap?
3、Why is the absorption efficiency improved more significantly in countercurrent operation?
4、Why does the deviation increase with increasing CO2 concentration in Figure 8? Is it possible to correct the model to extend the generalizability? Is the model applicable to other absorbers?
5、Why do the authors select the 3 kinds of CO2 concentrations (30%, 35% and 40%)? Where do these concentrations generally occur?
Author Response

(The authors gave the same response as above.)
